# Positive and Negative Emotion Regulation in Adolescence: Links to Anxiety and Depression

**DOI:** 10.3390/brainsci9040076

**Published:** 2019-03-29

**Authors:** Katherine S. Young, Christina F. Sandman, Michelle G. Craske

**Affiliations:** 1Social, Genetic and Development Psychiatry (SGDP) Centre, Institute of Psychology, Psychiatry and Neuroscience, King’s College, London SE5 8AF, UK; 2Department of Psychology, University of California, Los Angeles (UCLA), Los Angeles, CA 90095, USA; chrissysandman@ucla.edu (C.F.S.); MCraske@mednet.ucla.edu (M.G.C.); 3Department of Psychiatry and Biobehavioral Sciences, University of California, Los Angeles (UCLA), Los Angeles, CA 90095, USA

**Keywords:** anxiety, depression, adolescence, emotion regulation, fMRI, psychophysiology, psychological treatment

## Abstract

Emotion regulation skills develop substantially across adolescence, a period characterized by emotional challenges and developing regulatory neural circuitry. Adolescence is also a risk period for the new onset of anxiety and depressive disorders, psychopathologies which have long been associated with disruptions in regulation of positive and negative emotions. This paper reviews the current understanding of the role of disrupted emotion regulation in adolescent anxiety and depression, describing findings from self-report, behavioral, peripheral psychophysiological, and neural measures. Self-report studies robustly identified associations between emotion dysregulation and adolescent anxiety and depression. Findings from behavioral and psychophysiological studies are mixed, with some suggestion of specific impairments in reappraisal in anxiety. Results from neuroimaging studies broadly implicate altered functioning of amygdala-prefrontal cortical circuitries, although again, findings are mixed regarding specific patterns of altered neural functioning. Future work may benefit from focusing on designs that contrast effects of specific regulatory strategies, and isolate changes in emotional regulation from emotional reactivity. Approaches to improve treatments based on empirical evidence of disrupted emotion regulation in adolescents are also discussed. Future intervention studies might consider training and measurement of specific strategies in adolescents to better understand the role of emotion regulation as a treatment mechanism.

## 1. Introduction

Emotion regulation is defined broadly as the capacity to manage one’s own emotional responses. This includes strategies to increase, maintain, or decrease the intensity, duration, and trajectory of positive and negative emotions [1,2,3]. Learning to regulate emotions is a key socio-emotional skill that allows flexibility in emotionally-evocative situations. There are clear developmental shifts in how we manage emotional responses. In early childhood, emotions are frequently expressed and external support is sought (e.g., from a caregiver [4]). In adolescence, there is typically a decreased reliance on parental support and limited efficacy of adaptive internal emotion regulation [5]. As individuals mature into adulthood, emotional experiences are increasingly effectively managed through internal regulatory strategies [6]. Disruptions to emotion regulation capacities in adulthood are central to theories of how anxiety and depressive disorders manifest and are maintained [7,8]. These theories suggest that reduced capacities to downregulate heightened negative affect are common to both anxiety and depression, whereas reduced ability to regulate positive affect may be more specific to depressive disorders [9]. Many psychological interventions for anxiety and depression include cognitive or behavioral strategies that aim to improve abilities to regulate emotion [10,11].

Emotion regulation capacities develop substantially across adolescence. Studies of typically developing individuals suggest limited efficacy of internal regulatory strategies in early adolescence, shifting towards increased use of adaptive strategies and decreased use of maladaptive strategies with age [5,12]. This development coincides with changes in social environment and brain structure. Adolescence is a period of life with various emotional challenges, such as new academic or work-place pressures, increasing importance of peer and romantic relationships, and reduced dependence on family support [13]. Heightened emotional reactivity, increased risk-taking, and impulsive behaviors are also characteristic of adolescence [14]. This is coupled with ongoing neurobiological development among circuitries implicated in the management of emotional processes (for a review, see [15]). Investigation of normative development is ongoing, but current theories focus around maturation in activity and connectivity among the prefrontal cortex, striatum and amygdala across adolescence [16,17]. These models propose that increasing prefrontal control over emotionally reactive subcortical regions enhances capacities to regulate negative emotions (particularly fear) and manage impulsive tendencies (reward and approach [15,16,18]). 

Adolescence is a period of heightened risk for the onset of anxiety disorders and depression [19,20]. It is well-established that stressful life events and childhood adversity are substantial risk factors for future psychopathology [21]. There is also evidence suggesting that the capacity to regulate emotional reactions to these events may play a mediating role [22,23]. Given increased independence and novel demands during adolescence relative to childhood, adolescents may have a particular need to regulate their emotions in response to stressors. Failure to do so may confer risk for mental health problems. Thus, emotion regulation may be one important piece of a complex puzzle in terms of risk for anxiety and depression. The current paper addresses the evidence linking disrupted emotion regulation to the development of anxiety and depression in adolescence. This question has been investigated across different levels of analysis including self-report, behavioral, peripheral psychophysiological and neural measures. Repeated observations across multiple levels of analysis increase the reliability and validity of observed associations and may improve precision in understanding dysfunction and disease course [24,25]. Here we review the consistency of evidence across multiple modalities and highlight discrepancies and gaps in the literature.

A major challenge in the study of emotion regulation is definition and operationalization of the construct. In this review, we focus on evidence from the most widely-used measures of emotion regulation, rather than providing an exhaustive list of all possible measures. We begin with an overview of methodological approaches to studying emotion regulation most frequently used in adolescents. We then review evidence across levels of analysis supporting claims of a link between negative and positive emotion regulation capacities with anxiety and depression (summarized in Table 1; note that as a narrative rather than a systematic review, we provide a selection of findings of interest, rather than an exhaustive list of all findings in this area). Next, we discuss how these findings have informed current and emerging interventions targeting emotion regulation and their potential for adolescent populations. Finally, we provide an overview of discrepancies and gaps in current research and directions for future work that may enhance our understanding of the development of emotion (dys)regulation among adolescents at risk for anxiety and depression.

## 2. Overview of Measures of Emotion Regulation

Theoretical models of emotion regulation provide organizational frameworks within which to assess different strategies for regulation. The most widely used framework is the ‘process model of emotion regulation’ [2,11] which differentiates strategies along the timeline of a developing (negative) emotional response. A basic distinction in this model is between: (1) antecedent-focused strategies that manage the generation of an emotional reaction before it occurs, and (2) response-focused strategies that are invoked during an ongoing emotional reaction. A common antecedent-focused strategy is cognitive reappraisal, the process by which individuals consider a situation in a different way with the goal of managing their response when faced with that situation (e.g., when waiting for a friend to return a message, thinking ‘they are busy’ rather than thinking ‘they don’t like me’). Reappraisal is considered an adaptive regulatory strategy. A common response-focused strategy is expressive suppression, whereby individuals try to reduce or ‘suppress’ facial, vocal, or other expressions of the emotions they are currently experiencing. Expressive suppression is considered to be a maladaptive regulatory strategy. There are also numerous other strategies that impact the duration and intensity of negative emotions, such as problem solving, acceptance (considered to be adaptive) and rumination (maladaptive). Cognitive strategies for the regulation of positive emotions are not as widely discussed, but some focus on ‘enhancing’ and ‘dampening’, often in the context of interpersonal regulation between parents and children. Enhancing describes parental reactions of enthusiasm, encouragement or validation, whereas dampening refers to a focus on potential negative aspects of a situation, raising concerns and minimizing positive aspects [69].

## 3. Self-Report Measures of Emotion Regulation

A widely used self-report measure is the Emotion Regulation Questionnaire (ERQ) that follows the organizational principles of the process model of emotion regulation and has subscales for reappraisal and expressive suppression [70]. Other questionnaires assess different combinations of emotion regulation strategies, such as the Difficulties in Emotion Regulation Scale (DERS; [71]), the Cognitive Emotion Regulation Questionnaire (CERQ; [72]) and the *Fragebogen zur Erhebung der Emotionsregulation bei Kindern und Jugenlichen* (FEEL-KJ [73]). The varying content of these widely used self-report measures highlights inconsistencies with which the term ‘emotion regulation’ is used and limits the extent to which data across studies can be combined (see Table 2 for subscale comparison across measures). There are fewer standardized self-report measures available for positive emotion regulation. One such measure is the ‘responses to positive affect’ scale, which consists of three sub-scales: dampening, self-focused positive rumination and emotion-focused positive rumination [74]. 

Retrospective self-report questionnaires are criticized for the likelihood of over-generalized responding, the assumption that people are conscious of how they regulate their emotions and bias in memory effects (remembering most recent and/or salient experiences [75]; Table 3 lists methodological limitations of techniques discussed). Overcoming limitations based on memory, experience sampling methodologies aim to capture responses to experiences during, or close in time to, real life events through high density self-reporting (multiple times per day). This approach offers richer data on emotional experiences and often encompasses both positive and negative affect. Existing studies using this approach assess emotion regulation through self-report of strategy use and duration of emotional experiences (i.e., ‘emotional recovery’). However, ‘emotional recovery’ may be influenced by factors other than regulation, including emotional intensity or situational changes. This type of approach therefore prevents discrete measurement of emotional *reactivity* from emotional *regulation* (for a theoretical discussion of this issue, see [76]).

## 4. Behavioral Assessment of Emotion Regulation

Observational approaches can be used to examine responses, for example: during in vivo stress inductions (e.g., Trier Social Stress Task [77] or mock job interviews [46,59]); between pairs of individuals in spontaneous interactions; or during prescribed stress-inducing or rewarding situations [65,66,78]. Participants’ behavioral and verbal responses are coded and classified according to regulatory strategy and subjective affect ratings can be collected to measure emotional recovery. As with self-report measures, these approaches too may be influenced by socially desirable responding and lack the capacity to separate reactivity from regulation.

Computer-based methods of assessing emotion regulation behaviors involve presenting participants with affectively evocative images (such as from the International Affective Picture System [79]) and asking them to rate the strength of their emotional reaction. In some variants, participants passively view images to assess ‘automatic’ or ‘spontaneous’ regulation, other variants aim to enhance ecological validity by swapping affective images for descriptions of ambiguous situations (e.g., mother is late to come home [40]). While providing a degree of experimental control unavailable in observational studies, these ‘spontaneous regulation’ paradigms still cannot dissociate emotional reactivity from regulation, conflating assessment of the strength of an emotional response with the ability to regulate this response. Stronger ‘deliberate regulation’ designs compare ratings from passive ‘reactivity’ trials with active ‘regulatory’ trials in which participants are instructed to down- or up-regulate their emotional response. This approach offers a within-subjects inspection of the impact of deliberate emotion regulation using predetermined strategies. A potential drawback of this approach, however, is response bias in affect ratings where individuals may report reduced negative affect as a consequence of following task instructions rather than successful regulation per se. In some studies participants are trained to use specific strategies (such as reappraisal, distancing or suppression), although there is variability in the extensiveness of pre-task training and participant proficiency in strategy usage across studies.

## 5. Peripheral Psychophysiological Indicators of Emotion Regulation

Peripheral psychophysiological studies of emotion regulation use similar designs to those described above for behavioral assessments, so the limitations of those designs also apply to methods described here. Investigation of peripheral psychophysiological correlates of emotion regulation encompass a range of measures. Cardiac and respiratory measures include heart-rate variability and respiratory sinus arrhythmia (RSA; variation in heart rate within a breath cycle). Greater variation in heart rate and RSA are considered indicative of greater physiological adaptation to emotional stimuli (i.e., more effective regulation [80]). Ocular measures include pupil dilation, a measure of arousal or ‘cognitive effort’ [81], and visual fixation patterns, which demonstrate areas of attentional focus (and have also been suggested to indicate prefrontal cortex activation [82]). Facial electromyography (EMG) of the startle blink reflex and corrugator muscle activation are used as measures of negative emotional arousal [83]. Skin conductance levels and responses are used as a measure of emotional arousal at a chronic, or stimulus-evoked level, respectively [84]. These measures offer the potential for objective, low-cost biological markers of emotion regulation. However, they largely suffer from a lack of specificity in relation to psychological constructs, making the functional significance of differences observed difficult to interpret [85]. 

## 6. Neural Measures of Emotion Regulation

Neuroimaging studies using functional Magnetic Resonance Imaging (fMRI) to investigate neural correlates of emotion regulation have primarily used deliberate regulation paradigms. Across studies to date, instructions for regulation vary from broad approaches (e.g., ‘decrease’) to specific strategies (e.g., ‘distance’ or ‘reappraise’). One concern with fMRI designs is that due to timing constraints, participants are often given a short period of time (approximately eight seconds) to implement a strategy per image, raising potential concerns of ecological validity. A different approach used in fMRI studies are measures of ‘incidental regulation’, such as affect labeling [86,87]. Unlike study designs assuming ‘automatic’ regulation, studies of ‘incidental’ regulation investigate processes wherein a specific task may lead to emotion regulation, without the deliberate intention of doing so. For example, in the affect labeling task, participants view images of emotional facial expressions and are asked to label the emotion they see. Affect labeling has been shown to decrease experienced negative emotions and is also common across forms of psychotherapy [86]. As many individuals may be unaware of their emotion regulation strategies, the incidental nature of affect labeling may be helpful in addressing/circumventing limitations of self-report methodologies. One criticism of this approach, however, is that individuals do not intend to regulate when labeling (i.e., the goal is implicit), so it may not be considered a true form of emotion regulation. Despite this concern, studies using this task have demonstrated that affect labeling recruits neural circuitries implicated in emotion regulation in healthy adults, such as reduced amygdala activation, and increased inverse ventrolateral prefrontal cortex (vlPFC)–amygdala connectivity [86,87]. Because affect labeling robustly activates this circuitry in healthy samples, it offers an objective comparison of potential biological differences in psychopathology during incidental emotion regulation.

## 7. Relationships between Emotion Regulation Abilities and Symptoms of Anxiety and Depression in Adolescents

### 7.1. Findings from Self-Report Studies

Analyses of self-reported data consistently identify associations between emotion regulation abilities and symptoms of anxiety and depression in adolescents. For example, less use of cognitive reappraisal and greater use of expressive suppression was associated with higher symptoms of depression [46,48], and higher levels of rumination were associated with greater symptoms of social anxiety [39]. This was recently confirmed in a meta-analysis of 35 studies in adolescents (aged 13–18 years), demonstrating that compared to healthy individuals, those with anxiety and depressive disorders engaged in less reappraisal, problem solving, and acceptance (adaptive regulatory strategies) and more avoidance, suppression and rumination (maladaptive strategies [36]). Of these associations, the strongest effects were observed for reduced acceptance and increased avoidance and rumination across both anxiety and depression, with little evidence of specific disruptions linked to either disorder. Other work has sought to investigate patterns of disrupted emotion regulation specific to individual anxiety disorders. One study suggested greater deficits in emotional clarity and non-acceptance of emotions in social anxiety disorder compared to generalized anxiety disorder (using the DERS [38]). However, another found no differences between groups of adolescents with different anxiety disorder diagnoses (using the FEEL-KJ [37]). While use of different questionnaires across studies may explain differences in effects observed, there is no strong evidence of specific deficits in emotion regulation resulting in specific symptom profiles within anxiety disorders. 

Relatively few studies have examined the role of positive emotion regulation in relation to symptoms of anxiety and depression. One study in which parents reported on adolescent affect found that parents of depressed adolescents rated shorter durations of ‘happy’ affect in their children, compared to parents of non-depressed adolescents [61]. However, this study did not investigate strategies for maintaining or dampening positive affect, limiting the ability to differentiate disruptions in regulation from reactivity. Other studies have focused on interpersonal aspects of emotion regulation, showing that self-reported parental dampening, or a lack of parental enhancing of positive affect, was related to prospective increases in adolescent depression, potentially via their own dampening of positive affect [63,64]. The extent to which these findings are specific to symptoms of depression, rather than more general psychopathology remains unexplored.

Studies using experience sampling methodologies are beginning to examine relationships between daily experiences of emotion, regulation strategies and symptoms of anxiety and depression. One such study in adolescents aged 13–16 years over a 21-day period showed that symptoms of depression were related to reduced *variance* in reported emotional state (including happiness, depression, anger and anxiety), an effect that was associated with the ‘acceptance’ subscale of the DERS [47]. A study that collected data over two weekends using nine daily self-reports of emotional events and self-rated emotion regulation in a sample of 12–17 year-olds found no association between momentary use of emotion regulation strategies and depression in girls, but an inverse relationship between acceptance and depression in boys [88]. These types of approach hold much promise for examining daily life experiences of emotion regulation, but further work is required to standardize analytic approaches and investigate other factors that may influence these relationships potentially explaining the mixed findings observed to date.

Beyond simple correlations of co-occurring emotion regulation deficits and symptoms of anxiety and depression, it has been suggested that disrupted emotion regulation is a risk factor for the development of psychopathology [89,90]. Confirming this effect, meta-analytic data suggests that disrupted self-reported emotion regulation abilities predict subsequent diagnosis of anxiety or depression [36]. Critically, the same analyses did not find that psychopathology predicted subsequent disruptions to emotion regulation. This unidirectional relationship was also observed in a large (*N* = 1065) study of adolescents aged 11–14 years [75]. Although this study showed that while a latent construct of ‘emotion dysregulation’ (combining multiple subscales) predicted symptoms of anxiety, aggression and disordered eating behaviors, depression was predicted only by rumination, expression of anger and expression of sadness. While highlighting the differing effects that can be observed with varying definitions of emotion regulation, this work does provide support for the notion that disrupted emotion regulation is a risk factor for future psychopathology.

Emotion regulation has also been proposed as a mediating variable between a risk factor (e.g., early life adversity) and the development of psychopathology. Mediator variables hold the potential to identify factors that might be altered through intervention to reduce the risk of psychopathology. Studies investigating the mediating role of emotion regulation in adolescents suggest that increased use of maladaptive emotion regulation strategies may mediate the association between adversity and psychopathology. These studies found: (1) an effect of self-blame, catastrophizing, and rumination on the relationship between stressful life events and symptoms of depression [57]; and (2) a role for rumination and impulsive responding on the relationship between childhood maltreatment and symptoms of internalizing psychopathology [58]. What these studies do not indicate is whether higher levels of adaptive strategies reduce the risk of psychopathology following early life adversity.

### 7.2. Findings from Behavioral Studies

Across studies using deliberate emotion regulation paradigms, there is some evidence suggesting that anxiety is associated with reduced use of reappraisal, while findings for depression are mixed. Considering first anxiety, anxious adolescents were shown to have heightened emotional reactivity to negative images and impairments in generating reappraisals when cued [40,41]. However, in trials where they did successfully generate reappraisals, anxious adolescents were able to effectively reduce their negative affect to a similar degree as their non-anxious counterparts. Deficits in reappraisal generation corresponded both with less frequent self-reported everyday use of reappraisal and lower reappraisal self-efficacy (i.e., the belief that reappraisal would improve their feelings), suggesting a combination of real and perceived deficits in adaptive emotion regulation. Classification of anxious adolescents’ verbal responses to ambiguous situations according to regulatory strategy showed reduced spontaneous use of reappraisal and problem solving and increased use avoidance and help-seeking strategies with no differences in attentional deployment (distraction) or behavioral response modulation (suppression) [40]. Together, these findings suggest that reappraisal may be an effective yet underutilized strategy in adolescents with anxiety. Another study assessing abilities to suppress or amplify expressive behaviors in response to positive and negative images found no effects of anxiety or depression [42], suggesting impairments observed may be specific to reappraisal skills. 

Studies investigating reappraisal ability have demonstrated mixed effects for adolescent depression. All studies to date with behavioral affect rating data have used deliberate emotion regulation paradigms while participants also underwent fMRI (neural results are discussed below). Two studies found no difference in reappraisal success (difference in average affect ratings for ‘look’ minus ‘decrease’ trials in samples aged 13–17 [49] and aged 15 [50]), but a third study in a sample of 15–25 year-olds showed poorer reappraisal success in adolescents with depression compared to healthy controls [51]. One difference between studies that may contribute to the discrepancy in findings is depression severity, with deficits in reappraisal observed in a sample with more severe depressive symptoms (and an older age range). It is possible that depression severity impedes the effectiveness of reappraisal, although it remains unclear whether this is due to deficits in reappraisal generation or implementation. Future work directly comparing these processes in adolescents with anxiety, depression and mixed diagnoses would be helpful to delineate the nature of any differences associated with specific disorders.

Behavioral studies of interpersonal positive emotion regulation highlight an association between depressive symptoms and shorter duration of positive affect. In one study, adolescents and their parents completed a trivia game which was rigged to provide positive feedback, followed by a ‘conflict task’ in which families discussed previously identified ‘family issues’ [65]. While there was no association between depressive symptoms and observed positive affect during the reward task, there was an association with the ‘persistence’ of positive affect, defined as the maintenance of positive affect in a negative situation. Other findings suggest links between parental and adolescent emotion regulation, with reduced maternal positivity and increased dampening related to reduced maintenance of positive affect [66] and higher adolescent depressive symptomatology [67]. However, these studies did not assess self-focused regulatory strategies that may contribute to positive affect persistence, again impacting the ability to dissociate disruptions to emotion regulation from disrupted emotional reactivity.

As suggested in the self-report literature, there is emerging behavioral evidence that emotion regulation may impact the association between stressful experiences and psychopathology. In one study examining this effect, adolescents completed a social stress task (a mock job interview), provided distress ratings before and after the task, and completed self-report measures of cognitive reappraisal (using the ERQ) and depressive symptoms. Among those reporting higher levels of depressive symptoms, greater self-reported tendency to use cognitive reappraisal was associated with faster ‘emotional recovery’ (difference in distress ratings from before to 30 min after the task [59]). These findings indirectly suggest that the ability to use cognitive reappraisal in the face of social stressors may buffer the impact of depressive symptoms on emotional reactivity and recovery. However, it is important to note that in-vivo emotion regulation was not directly assessed during the stress task. Findings from self-report studies suggest that emotion regulation following stressful life events may impact the likelihood of developing psychopathology, while this study suggests a relationship in the opposite direction, that self-reported general emotion regulation tendencies may affect the impact of depression upon emotional reactivity/recovery. A variant on this paradigm in which participants are instructed to use reappraisal or other specific strategies in different conditions would allow a more direct investigation of the efficacy of each strategy. This would also help clarify the direction of these relationships, which would be useful in improving understanding of the developmental etiology of depression in adolescents (e.g., clarifying emotional reactivity or recovery as a vulnerability factor or a symptom of depression).

### 7.3. Findings from Studies of Peripheral Psychophysiology

Studies of peripheral psychophysiological indicators of emotion regulation in adolescents have sought to identify specific patterns of disruption linked to anxiety and depression. One small study in anxious youth (*N* = 27, aged 8–17 years) demonstrated that the number of fixations during negatively and positively valenced pictures (relative to neutral pictures) was greater among individuals with anxiety disorders, compared to healthy individuals [43]. The authors suggest that as visual fixations have been shown to correlate with activation of the prefrontal cortex [82], these findings may indicate that anxious adolescents were trying to regulate their responses even in the absence of instructions to do so. However, given that there were no differences in visual fixations when participants were instructed to regulate, this interpretation seems unlikely. The same study also found greater pupil dilation during negative compared to neutral pictures when instructed to ‘upregulate’ emotional responses in anxious, but not healthy adolescents [43]. As pupil dilation is considered an index of arousal this might suggest anxious adolescents experience more intense emotions when deliberately upregulating. However, the general nature of instructions prevents conclusions regarding the type of strategy employed, or whether, for example, adolescents with anxiety engage maladaptive regulatory strategies more readily than healthy adolescents. In addition, it is important to note that using psychophysiological measures to make inferences about emotional states is a form of ‘reverse inference’, and the lack of specificity between emotional experiences and peripheral psychophysiological markers caution against this type of conclusion. 

Whereas research in children suggests a predictive relationship between RSA and future anxiety and depression (e.g., [54,91]), findings from studies in adolescents are mixed. A large study of 11-year-olds (*N* = 1653) found no correlation between RSA and concurrent or future depressive symptoms assessed at age 13 [55]. The same study did, however, identify an interaction with life stress such that among individuals who experienced higher levels of stressful life events, higher RSA was associated with reduced self-reported anxiety [55]. Other work has suggested that atypical RSA patterns (either higher or lower) are associated with maladaptive regulatory strategies, which in turn are predictive of future depressive episodes in older adolescents with a history of depression (although RSA did not directly predict depression recurrence [53]). In another study, change over time in RSA predicted emotion regulation abilities in a sample of 8-12 year-olds with varying levels of depression and conduct problems [52]. Improving physiological responses to emotional challenges over time, i.e., increased RSA during a sad mood induction, was associated with fewer self-reported difficulties in emotion regulation, particularly in relation to ‘accepting’, ‘impulse control’ and ‘ability to use emotion regulation strategies’. RSA is often considered a specific measure of emotion regulation, yet it has also been shown to vary according to individual differences in emotional reactivity [80] which limits interpretations that can be made with this measure.

Investigating the effects of sustaining positive affect, a study of young adults (18–21 year-olds) involved a reward task, followed by a mood induction film clip that was positive, negative or neutral [92]. Reporting higher positive emotion during the reward task was associated with a faster return to physiological baseline (based on heart rate measures) when subsequently viewing a neutral film clip, but slower return to baseline when subsequently viewing a positive film clip. This study suggests that individual differences in reactivity to reward are related to physiological differences in adaptation to subsequent mood induction stimuli to maintain (positive clip) or reduce (neutral clip) positive affective states. However, this study did not examine the impact of intentional regulation of positive affect, which would be of much interest to investigate whether individuals can generate a ‘sustained’ positive valence state, and how this relates to symptoms of psychopathology. 

Overall, evidence from psychophysiological studies linking emotion dysregulation to anxiety and depression is preliminary and highly varied in experimental methodology and sample characteristics, making comparisons across studies difficult. Addressing some of these challenges, a recent study concurrently used a range of measures to assess emotion reactivity and regulation during presentation of valenced images [44]. In a sample of young adolescents, measures of corrugator and startle EMG and skin conductance were assessed while participants were instructed to ‘maintain’ or ‘discontinue’ their emotional responses. Corrugator EMG activity was sensitive to valence (positive vs. negative stimuli), while startle EMG and skin conductance was sensitive to regulation instruction. This approach offers promise for identifying reliable indicators of emotion regulation across development and how they may be disrupted in adolescent anxiety and depression.

### 7.4. Findings from fMRI Studies

In normative adolescent neural development, the maturation of prefrontal regions supporting emotion regulation lags behind limbic regions involved in emotion generation (for a review, see [15]). Most studies observe linear decreases in amygdala reactivity to affective stimuli with age [30,31,32], alongside linear increases in dorsomedial prefrontal cortex (dmPFC) recruitment [33]. Age-related improvements in cognitive regulation of emotion are also associated with reduced amygdala activation [34,93] as well as increases in inverse coupling (i.e., negative correlation in functional connectivity) between the PFC and amygdala [34]. A shift from positive to inverse amygdala–PFC connectivity occurs from childhood to adolescence [30]. By mid-adolescence, most youth display inverse amygdala–PFC connectivity, with stronger inverse connectivity corresponding to lower symptoms of anxiety in a non-clinical sample [30,94]. Evidence across fMRI studies suggests that disruptions in the same cortico-limbic circuitry during emotion regulation are implicated in anxiety and depression in adolescents.

Studies reviewed above described mixed findings as to whether adolescents with depression demonstrated disrupted abilities to reduce ratings of negative affect during instructed reappraisal [49,50,51]. In contrast, functional MRI data from the same studies have consistently found evidence of aberrant prefrontal activation and connectivity during deliberate emotion regulation. However, the specific regions implicated and disruptions in connectivity observed vary across studies (see Figure 1). Three out of four extant studies found evidence of heightened amygdala reactivity or greater amygdala–PFC connectivity during regulation in adolescents with depression [50,51,56]. However, the study which did not observe these findings was the largest (with the greatest power to detect effects) and did not find robust evidence of altered amygdala reactivity or connectivity, instead demonstrating changes in connectivity between dorsal regions of prefrontal cortex and inferior frontal regions [49]. Studies of depression in adults generally support a model of heightened activation in cognitive control regions and impaired subcortical down-regulation [95,96,97,98], which has been interpreted as an effortful yet ineffective attempt to regulate. In adolescents, connectivity between subregions of PFC may also play a role. Further investigation of inconsistencies across studies, perhaps by utilizing measures of emotion regulation from other levels of analysis, would be useful in determining the role of regulatory circuitry in adolescent depression. Importantly, no studies have investigated neural differences in up-regulation of positive emotions in adolescents with depression, which is an important avenue for future research given the relevance of the positive affect system in major depressive disorder.

fMRI investigations of emotion regulation in adolescents with anxiety have been more limited than in depression, with no studies of deliberate reappraisal to date. However, activation across similar circuitry during incidental emotion regulation may prospectively predict the development of anxiety symptoms. For example, in a sample of ninth grade females (mean age 15), positive amygdala–vlPFC connectivity during an incidental emotion regulation task (affect labeling [87]) predicted future symptoms of anxiety in the following 9 months [45]. Interestingly, childhood negative emotionality (assessed by parent, teacher, and self-report from grades 2–7) related to positive amygdala–right vlPFC connectivity in ninth grade, but only in girls with low levels of cognitive control (assessed by Brief Rating Inventory of Executive Functioning reports from grades 5–7). The authors suggested that individual differences in negative emotionality and cognitive control may be respective risk or resilience factors for ‘less mature’ positive connectivity between the amygdala and PFC, and in turn anxiety. 

Similar to behavioral and self-reported findings, there is some evidence that neural measures of emotion regulation ability (cortico-limbic functional connectivity) may influence the association between stress and depression. After completing an incidental emotion regulation task [87], a sample of adolescent females (mean age 15) underwent a social stress manipulation by completing the Cyberball task [99] in which they were unknowingly excluded from a virtual game of catch. Positive amygdala-vlPFC connectivity during incidental emotion regulation was associated with greater self-reported ‘stress-reactive rumination’ (following the Cyberball task) and mediated the relationship between self-reported rumination and depressive symptoms [60]. The retrospective self-report of depression and lack of temporal precedence limits these findings from a developmental psychopathology perspective, but highlights a potential mediating mechanism that could be investigated longitudinally in future research.

Taken together, these studies suggest that adolescents with anxiety and depression exhibit differences in neural functioning compared to non-depressed peers during deliberate emotion regulation. Evidence to date suggests that some of these differences may be similar to disruptions in emotion regulation neural circuitries observed in adults, although no studies have yet directly compared samples of adolescents and adults with anxiety or depression. Existing models of emotion regulation make inferences based on directional connectivity between prefrontal and subcortical brain regions. However, as functional connectivity analyses are correlational, it is ultimately impossible to interpret directionality (i.e., whether inverse connectivity indicates prefrontal down-regulation of affective regions). A less common yet promising analytical approach is ‘effective connectivity’ (e.g., dynamic causal modeling or Granger causality), which can be used to determine effective connectivity, or the directional influence of one region upon another. For example, one study using this technique demonstrated that adults with social anxiety disorder display impaired bidirectional amygdala–vmPFC effective connectivity while perceiving affective stimuli [100]. Use of this approach, and other advanced analytic techniques, may allow more direct investigation of proposed models of neural circuitry dysfunction during emotion regulation in adolescents with psychopathology.

## 8. Clinical Implications for Interventions in Adolescents

Understanding emotion regulation in adolescents with anxiety and depression is critical for improving the efficacy of existing treatments and informing the development of novel interventions. Promoting adaptive emotion regulation is a central component of most evidence-based psychotherapies for adolescent anxiety and depression, although different skills are emphasized across modalities. Cognitive Behavioral Therapy (CBT), emphasizes cognitive restructuring and promotes the use of reappraisal, while ‘third wave’ psychotherapies (e.g., mindfulness-based cognitive therapy, dialectical behavioral therapy [101]) focus on acceptance and decentering to regulate emotions. Most studies of psychotherapy effectiveness include both children and adolescents in combined samples, with age relating to better treatment outcomes [102]. Older adolescents may be better able to benefit from CBT possibly due to more developed cognitive and social skills, consistent with age-related improvements in emotion regulation ability in healthy adolescents [26,27]. It remains unknown whether emotion regulation skills taught in mindfulness-based versus cognitive behavioral approaches are better suited for certain individuals across development, highlighting the importance of age effects and treatment matching in future research.

As intervention packages typically contain several elements, it can be difficult to tease apart the ‘active ingredients’ of treatments. In line with the National Institute of Mental Health’s (NIMH) shift in clinical trials to an experimental therapeutic paradigm [103], a priority for intervention research is to test specific mechanisms of action that account for meaningful clinical change. Emotion regulation is a prime candidate for such mechanistic studies. This has been the goal in more recent treatments that specifically focus on enhancing emotion (e.g., Contextual Emotion Regulation Therapy [104], Emotion Regulation Therapy [105]). Changes in decentering and reappraisal through Emotion Regulation Therapy temporally preceded reductions in anxiety and depression in young adults, suggesting a potential mechanism [106]. Future work should extend and tailor these treatments to adolescent populations. 

Other mechanistic work aiming to distil the effects of individual treatment components has focused on briefer computerized trainings designed to change attentional or interpretational biases believed to contribute to anxiety and depression [107]. In line with the process model of emotion regulation [2], Cognitive Bias Modification aims to tap into antecedent-focused regulatory processes such as attentional deployment (Attention Bias Modification) and interpretation/reappraisal (Interpretation Bias Modification). Although these approaches have been shown to effectively retrain biases, estimates of the effects on clinical outcomes in adults are modest [108]. Recent adaptations that train attention toward positive stimuli show promise in reducing symptoms of anxiety and depression in children [109,110]. As reviewed above, adolescents with anxiety and depression may have specific deficits in the generation of reappraisals. More open-ended modifications of interpretation bias training may therefore be helpful in improving this ability. 

Few interventions target positive emotion regulation in adolescence and adults alike, mirroring the relative dearth of research in this domain. Designed specifically to treat anhedonia in adults, Positive Affect Treatment (PAT [111]) promotes positive emotion through a variety of behavioral, cognitive, and experiential exercises. For example, rather than challenging negative thoughts as in traditional CBT, PAT promotes identifying positive aspects of situations (i.e., finding the silver lining). Through its treatment components, PAT likely both induces and augments positive affect, involving both bottom-up and top-down processes (i.e., emotional reactivity and regulation). Future research might adapt similar interventions for adolescents. Given the link between adolescent depressive symptoms and reduced positive emotion persistence [65,66] novel interventions may focus on techniques that sustain positive affect in the presence of stress and train recovery after stressful events. 

Intervention studies also offer a powerful approach to investigating mechanisms of treatment action. Increasingly, neuroimaging measures have been included in trials of psychological interventions, with mounting evidence suggesting changes in functioning and connectivity in amygdala-prefrontal circuitry following CBT [112,113]. To date, there have been no studies of interventions with adolescents assessing neural mechanisms of interventions using emotion regulation tasks. Neuroscientific research of treatment mechanisms has started to lead to the development of novel treatment approaches, such as repeated transcranial magnetic stimulation (rTMS [114]) and neurofeedback [115,116], which hold promise for altering activation of emotion regulation neural circuitries.

## 9. Summary and Directions for Future Research

From the literature reviewed above (summarized in Table 1), there is a consistent body of evidence from self-report studies that disruptions to emotion regulation capacities are associated with greater likelihood of experiencing anxiety and depression in adolescence. There is also evidence suggesting that these disruptions to emotion regulation are predictive, rather than sequelae, of future psychopathology. To date, there is no strong evidence relating specific regulatory strategies with specific diagnoses or symptom profiles, suggesting that altered capacities in this domain confer a more general risk for psychopathology. 

In contrast, findings from behavioral studies suggest that anxiety in adolescence may be specifically related to a reduced spontaneous use of reappraisal regulatory strategies. However, given that there are far fewer behavioral than self-report studies in this domain and that behavioral studies have less comprehensively assessed all forms of emotion regulation across different diagnoses, the specificity of this effect may not be as clear as it appears. There is no consensus from behavioral research as to whether depression is linked to disruptions in regulation of negative affect, with some studies showing reduced reappraisal efficacy and others not showing this effect. One finding that does appear more consistent is the reduced duration of positive affect among adolescents with depression, although the extent to which this is tied to deficits in cognitive regulatory strategies has not been investigated. 

Findings from peripheral psychophysiological measures are limited, and effects observed are also mixed. There is some preliminary evidence from individual studies, but the variance in methods used prevents commentary on consensus of findings in this area. There has been a relative proliferation of functional MRI studies assessing disrupted emotion regulation neural circuitry. On the whole, these studies have identified differences in activation and functional connectivity between amygdala and prefrontal cortical brain regions in adolescents with depression. Studies of anxiety suggest that disruptions in neural functioning may precede onset of symptomatology. Although overall findings from neuroimaging studies point to disruptions in similar circuitries, individual studies show different spatial patterns of effects. A challenge to future work in this area is to establish greater specificity in models of emotion regulation neural circuitry, including tests of effective connectivity that can begin to investigate probable direction of information flow.

Across studies of self-report, behavioral, and neural measures of emotion regulation reviewed, there were findings indicating relationships between reactivity to stressful events, emotion dysregulation and psychopathology. Findings from self-report studies suggest that emotion regulation skills may mediate the effects between early life adversity and subsequent psychopathology, while evidence from other levels of analysis present less clear directionality. It may be that disruptions to emotion reactivity and regulation are vulnerability factors for the development of future psychopathology, or that these problems arise as symptoms of specific disorders. 

### Future Directions

As demonstrated in Table 1, there are clear gaps in current research on associations between emotion regulation and psychopathology in adolescents. One particular discrepancy is the greater focus on regulation of negative emotions, compared to positive emotions. Both the theory and (self-report) measurement tools available are more established for negative compared to positive regulation. Approaches used to investigate regulatory skills in behavioral, psychophysiological and neural levels of analysis however, may be just as appropriate for the study of positive emotion regulation. Some of the studies reviewed used multiple techniques to investigate emotion regulation across different levels of analysis. This should be encouraged in future work, particularly in the integration of newer techniques, such as ecological momentary assessment, to allow investigation of how findings observed in retrospective self-report or lab-based studies relate to daily life experiences. 

As noted throughout, many of the studies reviewed also rely on indirect measures of emotion regulation, wherein responses to emotional stimuli are measured and the magnitude or duration of response is considered evidence of regulation. More stringent study designs use direct comparisons of instructed strategies which can help to disentangle effects of emotional *reactivity* from *regulation*. The instructions provided and regulatory strategies used in these studies is somewhat varied, but overall has focused on reappraisal, with less research investigating other regulatory skills (e.g., acceptance). Studies also vary in the use of emotional stimuli, but there has been a lack of discussion of whether there may be some strategies that are more appropriate than others for certain stimuli. For example, reappraisal may be an appropriate strategy for social stimuli, but less appropriate when responding to a moral violation (e.g., [27]).

There are also some individual difference variables that may be of much value to understanding the development of emotion regulation capacities. These include gender, pubertal status and cognitive abilities. Each of these have been suggested to impact the relationship between emotion regulation and psychopathology (e.g., [117,118,119]) and may be of interest in future work. Finally, further work investigating mechanisms of psychological interventions targeting emotion regulation abilities may be a particularly promising approach. This would allow a well-controlled investigation of whether training to enhance cognitive strategies for emotion regulation in adolescents mediates the impact of psychological therapies on symptoms of anxiety and depression.

## Figures and Tables

**Figure 1 brainsci-09-00076-f001:**
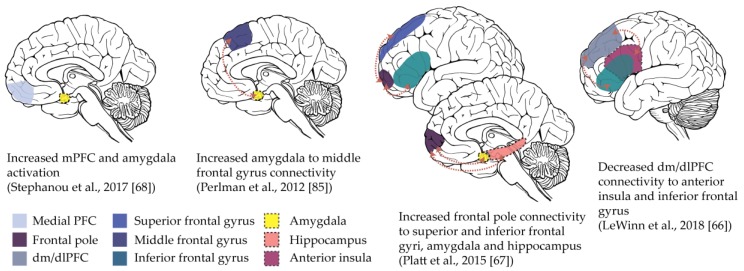
Patterns of altered neural activation and connectivity during emotion regulation in adolescents with depression. Overall, studies to date have demonstrated altered activation and connectivity in the amygdala and across regions of prefrontal cortex. The directionality of effects (greater or lesser in depressed compared to non-depressed participants), and the specific set of regions involved however varies across studies. (PFC: prefrontal cortex, dm/dlPFC: dorsomedial/dorsolateral PFC)

**Table 1 brainsci-09-00076-t001:** Reviewed evidence investigating links between emotion regulation and anxiety and depression in adolescence. Findings are organized according to negative and positive emotion regulation, and by methodology. (dlPFC: dorsolateral prefrontal cortex; dmPFC: dorsomedial prefrontal cortex; IFG: inferior frontal gyrus; IFL: inferior frontal lobule; MFG: middle frontal gyrus; PFC: prefrontal cortex; RSA: respiratory sinus arrhythmia; SFG: superior frontal gyrus; vlPFC: ventrolateral PFC).

Self-Report	Behavioral	Psychophysiological	Neural (fMRI)
**Normative Age-Related Changes**			
Increased use of ‘adaptive’ strategies, less use of ‘maladaptive’ strategies with age [5,12].	Reappraisal, but not distraction, improves linearly with age (ability does not always correlate with self-reported everyday use [26,27,28]).	Some evidence of age-related changes in RSA across adolescence [29].	Reduced amygdala reactivity with age [30,31,32,33], greater inverse PFC-amygdala connectivity, indicating better ‘top-down’ regulation [34,35].
**Negative Emotion Regulation**			
Associations with symptoms of anxiety			
More use of ‘maladaptive’ and less use of ‘adaptive’ strategies in anxiety disorders [36,37]. Social anxiety linked to reduced ‘emotional clarity’, reduced acceptance [38], and increased rumination [39].	Impaired reappraisal generation in anxiety disorders [40,41]. No differences in ‘amplifying’ or ‘suppressing’ expressive behaviors [42].	Greater number of visual fixations during negative images [43] and greater pupil dilation when ‘upregulating’ response to negative images [44] in adolescents with anxiety disorders.	Positive amygdala–vlPFC connectivity during affect labeling predicted future anxiety symptoms [45].
Associations with symptoms of depression			
More use of ‘maladaptive’, less use of ‘adaptive’ strategies in depression [36]. Specifically, less use of reappraisal [46], reduced acceptance [47] and higher suppression [48].	Mixed findings for reappraisal efficacy [49,50,51] in adolescents with depression.	Changes in RSA with age, linked to better ‘acceptance’, ‘impulse control’ and ‘ability to use emotion regulation strategies’ [52] in individuals with depression and conduct problems. RSA predicts more maladaptive emotion regulation in previously depressed adolescents [53]. Limited evidence of direct relationship between RSA and depression [54,55].	Evidence of disrupted activation and connectivity across emotion regulation neural circuitry (e.g., amygdala, PFC) in depression, but specific patterns of effects vary across studies ([49,50,51,56], see Figure 1).
Impacts link between stress and psychopathology			
Self-blame, catastrophizing, and rumination mediates the association between stress and depression [57]; rumination and impulsive responding links stress and internalizing symptoms [58].	Cognitive reappraisal mediates link between depressive symptoms and ‘emotional recovery’ from an experimental stressor [59].	RSA mediates the association between stress and anxiety [55]	Amygdala–vlPFC connectivity during incidental emotion regulation mediates the relationship between rumination and depressive symptoms [60]
**Positive Emotion Regulation**			
Associations with symptoms of anxiety			
Not investigated	Not investigated	Greater number of visual fixations during positive images in adolescents with anxiety disorders [43].	Not investigated
Associations with symptoms of depression			
Lower levels and shorter duration of positive affect [61,62], parental and self ‘dampening’ of positive emotions [63], lack of parental ‘enhancing’ [64] associated with depressive symptoms.	Reduced persistence of positive affect in conflict situation [65], low maternal positivity [66], and increased maternal dampening [67] associated with depressive symptoms.	Not investigated	Reduced activation of ventral striatum and PFC in response to reward (Forbes, 2011 #123 [68]), regulation not investigated
Impacts link between stress and psychopathology			
Not investigated	Not investigated	Not investigated	Not investigated

**Table 2 brainsci-09-00076-t002:** Overview of subscales across self-report measures of negative emotion regulation. Strategies are informally categorized as ‘adaptive’, ‘maladaptive’ or ‘uncategorized’ (describing more general emotion regulation behavior, rather than specific strategies). ERQ: Emotion Regulation Questionnaire; DERS: Difficulties in Emotion Regulation Scale; CERQ: Cognitive Emotion Regulations Questionnaire; FEEL-KJ: *Fragebogen zur Ehrebung her Emotionsregulation bei Kindern und Jugenlichen*.

ERQ	DERS	CERQ	FEEL-KJ
**Adaptive Strategies**			
Reappraisal		Positive reappraisal	Revaluation
	Non-acceptance	Acceptance	Acceptance
		Putting in perspective	
		Positive refocusing	
		Refocus on planning	
			Problem solving
			Cognitive problem solving
			Distraction
			Forgetting
			Humor enhancement
**Maladaptive Strategies**			
Expressive suppression			Emotional control
		Self-blame	Self-devaluation
		Other-blame	
		Rumination	Rumination
		Catastrophizing	
			Giving-up
**Uncategorized**			
	Goal-directed behavior		
	Impulse control		Aggressive actions
	Emotional awareness		
	Accessing regulation strategies		
	Emotional clarity		
			Withdrawal
			Social support
			Expression

**Table 3 brainsci-09-00076-t003:** Comparison of the methodological limitations of different study designs used to assess emotion regulation across levels of analysis. SR: self-report; Beh: behavioral; PP: peripheral psychophysiological; Neu: neural.

	SR: Questionnaire	SR: Experience Sampling	Beh: Stressful Situation	Beh: Observed Interactions	Beh/PP/Neu: Spontaneous Regulation	Beh/PP/Neu: Deliberate Regulation	Neu: Implicit Regulation
**Methodological Limitation**							
Varying content across measures	x			x			
Limited assessment of positive vs. negative affect	x		x		x	x	x
Retrospective bias	x						
Socially desirable responding	x	x	x	x		x	
Conflates emotional reactivity and regulation		x	x	x	x		
Assumes accurate insight into regulatory strategy	x	x					
Lacks ecological validity					x	x	x

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
