# Peer review of "Positive and Negative Emotion Regulation in Adolescence: Links to Anxiety and Depression"

_brainsci, 2019, doi:10.3390/brainsci9040076_

Round 1
Reviewer 1 Report
This timely review by Michelle Craske and colleagues has systematically summarized recent progress in the understanding of the link between emotion regulation and anxiety, depression in the adolescence. In particular, the authors focused on studies that assessed emotional regulation by self-report, behavioral, peripheral psychophysiological, and neural measures.
The general topic of this manuscript is attractive and should be of interest to a broad readership of Brain Science. While the literature cited and discussed is extensive, the organization of this article is smooth and concise. With that being said, I have the following concerns:
I encourage the authors to use illustrations and figures in the manuscript. They will convey the relevant message more directly and clearly. For instance, a figure could be potentially added to summarize the findings from fMRI studies, particularly for the different brain areas and their distinct role in emotional regulation.
Author Response
We thank the reviewer for this suggestion and have added a figure summarizing the fMRI findings as suggested. We have also added two further tables demonstrating: i) similarities and differences in emotion regulation subscales across self-report measures, and ii) comparison of methodological challenges across study designs employed at different levels of analysis. We hope that these additions help to clarify the main points of our review more effectively.
Reviewer 2 Report
This is a well-written review paper discussing the state of the literature on emotion regulation of positive and negative emotions in adolescence and the links to anxiety and depression. This is quite specific and has some novelty to it, although the general theme of the paper has been reviewed before (e.g., ER strategies in depression/anxiety disorders). There are several positive aspects to the review: it nicely outlines 4 areas of research (self-report, behavioral, physioloigical, and neuroimaing) and the authors ground themselves in the process model of emotion regulation. The literature review turns up some clearly under-researched areas which are important to consider.
I have some additional general/specific considerations that may help improve the paper for publication:
(1) There is no indication of how the "literature review" was conducted. Typically a methods section is included with some indication of what the search terms were and which databases. This is a big research area, some indication of how many articles were surveyed would be appropriate. It is unlikely that all papers were accessed. What are the search terms? How far did you go back into the literature and when did you stop searching? How many potential papers were persued?
(2) The paper is quite long, even for a review. The authors should aim to summarize and integrate the findings more, rather than talk about a specific study at length. In each section on the actual results from the literature search, a lot of space is given to individual studies. The authors are integrating from a number of different methods, which will clearly impact the reliability of the findings. The review could focus on a higher level analysis of the studies. Moreover, the authors have "front-loaded" a lot of the methods info by first describing the various methods and techniques that have been used across self-report, behavioral, physiological, and neuroimaging measures. I would run through the paper for succinctness and clarity, to ensure an easier read for readers and to stay focused on message (e.g., the state of the literature in each area).
(3) There is one part of Table 1 (and therefore the results section) that I think is under-represented by the literature available. Under "association with symptoms of depression" the category physiological, the authors list "no relationship to RSA". I honestly don't think this is true. At first, I thought possibly because I have seen a lot of adult research, but a quick Google Scholar search suggests that this has been studied in more detail than suggested (however, I used the search term "emotion dysregulation" to find this, leading me back to Point 1 about which search terms were used and when the literature was sampled/how). Specifically, Maria Kovacs has done a lot of work in this area (e.g., RSA and depression in young girls/adolescents). And I believe there are some significant predictive findings for depression in relation to resting RSA and another index called RSA reactivity.
e.g., Gentzler, A. L., Santucci, A. K., Kovacs, M., & Fox, N. A. (2009). Respiratory sinus arrhythmia reactivity predicts emotion regulation and depressive symptoms in at-risk and control children. Biological psychology, 82(2), 156-163.
e.g. Maladaptive mood repair, atypical respiratory sinus arrhythmia, and risk of a recurrent major depressive episode among adolescents with prior major depression
M Kovács, I Yaroslavsky, J Rottenberg, CJ George… - Psychological medicine, 2016
(4) The "interventions targeting emotion regulation in adolescence" section seems disconnected with the rest of the review and contains a lot of ancillary information not relevant to the current review. For example, the authors talk about the "lack of dismantling studies" (p.16, line 544) and rTMS later on (line 596). I just did not understand how this fits well with the rest of the findings above pertaining to the review of the studies. Here's where the authors could cut down on the manuscript. The section could be summarized more succinctly as a "clinical implications" section. Yes, ER is important in anxiety and depression and studies indicate that these change over the course of therapy/training. I don't think it's necessary to comment on all these different therapies and why they might be working.
(5) I would recommend abbreviating emotion regulation as ER, it just appears a lot of times and it will cut down on space.
The paper is generally well-written and I believe it to be meritful for publication once these additional notes are addressed. The topic is certainly relevant but I feel the authors may have covered too many topics in one article. But the focus is on methods and gaps in the current literature. Hence, why I think the focus should be on the methods and less on clinica implications at the end. The authors don't talk about clinical implications in the abstract anyways. I hope these suggestions have been helpful, having just written a major review article in a similar area myself.
Author Response
Below is a point-by-point response to the reviewer's comments.
(1) There is no indication of how the "literature review" was conducted. Typically a methods section is included with some indication of what the search terms were and which databases. This is a big research area, some indication of how many articles were surveyed would be appropriate. It is unlikely that all papers were accessed. What are the search terms? How far did you go back into the literature and when did you stop searching? How many potential papers were persued?
Response: As a narrative review, we did not conduct an exhaustive literature search, instead aiming to cover the some of the most commonly used measures and findings of most interest - we now explicitly state this in the manuscript: “note that as a narrative rather than a systematic review, we provide a selection of findings of interest, rather than an exhaustive list of all findings in this area”
(2) The paper is quite long, even for a review. The authors should aim to summarize and integrate the findings more, rather than talk about a specific study at length. In each section on the actual results from the literature search, a lot of space is given to individual studies. The authors are integrating from a number of different methods, which will clearly impact the reliability of the findings. The review could focus on a higher level analysis of the studies. Moreover, the authors have "front-loaded" a lot of the methods info by first describing the various methods and techniques that have been used across self-report, behavioral, physiological, and neuroimaging measures. I would run through the paper for succinctness and clarity, to ensure an easier read for readers and to stay focused on message (e.g., the state of the literature in each area).
Response: We have now gone over the entire paper again and found various places to summarize further, remove unnecessary detail and to make the work overall more succinct. Additionally, as suggested by reviewer 1, we have added tables and a figure to help communicate the main messages in this work more directly.
(3) There is one part of Table 1 (and therefore the results section) that I think is under-represented by the literature available. Under "association with symptoms of depression" the category physiological, the authors list "no relationship to RSA". I honestly don't think this is true. At first, I thought possibly because I have seen a lot of adult research, but a quick Google Scholar search suggests that this has been studied in more detail than suggested (however, I used the search term "emotion dysregulation" to find this, leading me back to Point 1 about which search terms were used and when the literature was sampled/how). Specifically, Maria Kovacs has done a lot of work in this area (e.g., RSA and depression in young girls/adolescents). And I believe there are some significant predictive findings for depression in relation to resting RSA and another index called RSA reactivity.
e.g., Gentzler, A. L., Santucci, A. K., Kovacs, M., & Fox, N. A. (2009). Respiratory sinus arrhythmia reactivity predicts emotion regulation and depressive symptoms in at-risk and control children. Biological psychology, 82(2), 156-163.
e.g. Maladaptive mood repair, atypical respiratory sinus arrhythmia, and risk of a recurrent major depressive episode among adolescents with prior major depression
M Kovács, I Yaroslavsky, J Rottenberg, CJ George… - Psychological medicine, 2016
Response: Thank you for drawing our attention to this area, and for the additional references provided. We too were surprised at the lack of evidence demonstrating this pattern in adolescents and have updated this section as follows:
We have not described the suggested Gentzler paper in detail as it seems from the methods section that all participants in the study were aged under 13 years. Throughout the paper, we have limited our review to studies of adolescents, rather than children or adults. We do, however, now include a sentence at the start of this paragraph to draw the reader’s attention to the discrepancy with the findings in children “Whereas research in children suggests a predictive relationship between RSA and future anxiety and depression [e.g., 71, 72], findings from studies in adolescents are mixed.”
The findings from the Kovacs paper do show a link between RSA and maladaptive emotion regulation, which in turn is linked with depression, although, like the other paper we cite in this section, do not find a direct association between RSA and depression. We have included description of this finding in this paragraph: “Other work has suggested that atypical RSA patterns (either higher or lower) are associated with maladaptive regulatory strategies, which in turn are predictive of future depressive episodes in older adolescents with a history of depression (although RSA did not directly predict depression recurrence [74]). ”
We acknowledge that our conclusion of ‘no relationship’ in the table is premature given the lack of literature available. We have updated the table to instead state findings related to ER strategies and also state: ‘Limited evidence of direct relationship between RSA and depression’
(4) The "interventions targeting emotion regulation in adolescence" section seems disconnected with the rest of the review and contains a lot of ancillary information not relevant to the current review. For example, the authors talk about the "lack of dismantling studies" (p.16, line 544) and rTMS later on (line 596). I just did not understand how this fits well with the rest of the findings above pertaining to the review of the studies. Here's where the authors could cut down on the manuscript. The section could be summarized more succinctly as a "clinical implications" section. Yes, ER is important in anxiety and depression and studies indicate that these change over the course of therapy/training. I don't think it's necessary to comment on all these different therapies and why they might be working.
Response: We thank the reviewer for this comment and pointing out where we can make this paper more succinct. We have substantially reduced this section, cutting a lot of the details on individual therapies. Our comments on dismantling studies and rTMS aimed to cover the range of approaches researchers are taking to try to leverage our understanding of emotion regulation into more targeted and effective interventions. We have also changed the title of this section and to make this clearer to the reader.
(5) I would recommend abbreviating emotion regulation as ER, it just appears a lot of times and it will cut down on space.
Response: We agree that it would be nice to abbreviate where possible - however, as we discuss both emotion regulation and emotion reactivity (and highlight how this distinction is not always clear in the available literature) we would prefer to keep things as is, to avoid any confusion in these terms.
The paper is generally well-written and I believe it to be meritful for publication once these additional notes are addressed. The topic is certainly relevant but I feel the authors may have covered too many topics in one article. But the focus is on methods and gaps in the current literature. Hence, why I think the focus should be on the methods and less on clinical implications at the end. The authors don't talk about clinical implications in the abstract anyways. I hope these suggestions have been helpful, having just written a major review article in a similar area myself.
Response: As described above, we have substantially reduced the clinical implications section to focus specifically on how advances through basic science can inform intervention research.
Reviewer 3 Report
Having read the manuscript entitled “Positive and negative emotion regulation in adolescence: links to anxiety and depression”, I have to admit that the topic of the submitted review is very interesting. The manuscript is well-written and very informative. However, I have several minor comments:
* In my opinion, length of the manuscript is too exhaustive and needs to be shortened
significantly. This can be achieved through incorporating tabular information instead of discussing in the text.
* I would definitely shorten the sections related to the measures of emotion regulation. Only the most important information should be given.
* Table 1 should be re-written. The reader may have difficulty in understanding information presented in the table without referring to the main text.
*Abbreviations used in the table should be explained below the table.
Author Response
* In my opinion, length of the manuscript is too exhaustive and needs to be shortened
significantly. This can be achieved through incorporating tabular information instead of discussing in the text.
Response: We now include 2 additional tables and a figure summarizing main points from the text. We have also gone through the manuscript and shortened/summarized/removed extra details where we could. We hope that this makes the work more digestible to the reader.
* I would definitely shorten the sections related to the measures of emotion regulation. Only the most important information should be given.
Response: We have reduced the content of this section considerably. We also summarize this section in 2 tables (one highlighting differences across questionnaire measures of emotion regulation and the other comparing methodological limitations across study designs at different levels of analysis). We hope this helps in communicating our method more directly, while still retaining sufficient detail in the text for the interested reader.
* Table 1 should be re-written. The reader may have difficulty in understanding information presented in the table without referring to the main text.
Response: We have re-written the table as suggested, hoping to provide enough explanatory text while still summarising findings in brief.
*Abbreviations used in the table should be explained below the table.
Response: We have added all abbreviations to the table caption
Reviewer 4 Report
The current review highlights age-related changes in emotion regulation with a focus on adolescence and links to anxiety and depression. Strengths of the review include the developmental framing, focus on positive and negative emotions, and integration across levels (e.g., self-report, behavioral, physiological, neural). I have minor suggestions that may help to refine the review.
The review motivates the focus on adolescence in part by pointing out that this is a period of increased risk for anxiety and depression, presumably relative to childhood and adulthood. Yet the age-related changes that are reviewed (e.g., self-report, behavioral, neural) are largely linear, i.e., linear improvements in emotion regulation. Thus, it is not immediately clear why adolescents would be more at risk than children, and this is a point that needs to be better reconciled in the text. It may be helpful to more explicitly state that emotion regulation is one piece of a complex picture in terms of risk for anxiety and depression.
Related to the age-related neural changes that are reviewed, it could be useful to note that most studies have observed linear decreases in amygdala reactivity with age (e.g., Decety, Michalska, Kinzler, 2012; Gee et al., 2013; Silvers et al., 2014; Swartz et al., 2014).
When the authors discuss inverse amygdala-PFC functional connectivity during adolescence, it would be helpful to cite Gee et al., 2013 JNeuro and Wu et al., 2016 HBM which show this normative developmental change (and Gee et al., 2013 also shows that the inverse functional connectivity is associated with lower symptoms of anxiety).
Author Response
The review motivates the focus on adolescence in part by pointing out that this is a period of increased risk for anxiety and depression, presumably relative to childhood and adulthood. Yet the age-related changes that are reviewed (e.g., self-report, behavioral, neural) are largely linear, i.e., linear improvements in emotion regulation. Thus, it is not immediately clear why adolescents would be more at risk than children, and this is a point that needs to be better reconciled in the text. It may be helpful to more explicitly state that emotion regulation is one piece of a complex picture in terms of risk for anxiety and depression.
Response: We have now aimed to clarify this in the introduction to make it clear why adolescence is a particularly important period for emotion regulation: “Given increased independence and novel demands during adolescence relative to childhood, adolescents may have a greater need to regulate their emotions in response to stressors. Failure to do so may confer risk for mental health problems. Thus, emotion regulation may be one important piece of a complex puzzle in terms of risk for anxiety and depression.”
Related to the age-related neural changes that are reviewed, it could be useful to note that most studies have observed linear decreases in amygdala reactivity with age (e.g., Decety, Michalska, Kinzler, 2012; Gee et al., 2013; Silvers et al., 2014; Swartz et al., 2014).
Response: We appreciate the additional references and agree it is important to highlight this point. We have cited several of these articles in the table (Decety et al., 2012; Gee et al., 2013; Swartz et al., 2014) and have added the following text to the section on fMRI findings: “Most studies observe linear decreases in amygdala reactivity to affective stimuli with age (86-88), alongside linear increases in dmPFC recruitment (89).“
When the authors discuss inverse amygdala-PFC functional connectivity during adolescence, it would be helpful to cite Gee et al., 2013 JNeuro and Wu et al., 2016 HBM which show this normative developmental change (and Gee et al., 2013 also shows that the inverse functional connectivity is associated with lower symptoms of anxiety).
Response: We thank the reviewer for highlighting an incorrect citation. We have replaced Gee et al., 2014 with Gee et al., 2013 and have added Wu et al., 2016. We have also clarified the shift in connectivity as it relates to symptoms: “A shift from positive to inverse amygdala-PFC connectivity occurs from childhood to adolescence [Gee et al., 2013]. By mid-adolescence, most youth display inverse amygdala-PFC connectivity, with stronger inverse connectivity corresponding to lower symptoms of anxiety in a non-clinical sample.”
Round 2
Reviewer 2 Report
I think the authors have attended well to the comments provided by 2 reviewers. They have shortened the paper and addressed my comments by adding statements to clarify their review as "narrative" as opposed to "systematic" and the concern about a statment made in one of the tables. The authors have also responded well to the suggested edits around the clinical implications section. Good work!